# Learning Slip with a Patterned Capacitive Tactile Sensor

Yuri Gloumakov, *Member*, *IEEE*, Tae Myung Huh, *Member*, *IEEE*, Hannah Stuart, *Member*, *IEEE*

*Abstract*— **The task of dynamically manipulating objects within a robotic hand presents ongoing challenges. In particular, friction and slip often dictate task success yet remain difficult to measure directly, quickly, and accurately; this includes both the detection of slip events and slip speed. Complex solutions exist that involve training a control policy using neural networks, with image-based sensors or external cameras, or when contact geometry can be inferred. Using only a capacitive sensor with a `nib`-patterned structure, we attempt to demonstrate the sensor's ability to detect slip speed during uninterrupted contact where geometry cannot be inferred, while benefitting from faster sensing, cheaper construction, and smaller profile. We hope that by collecting vibration amplitude and frequency and applying supervised learning techniques to directly measure slip speed we can guide an implementation of manipulation controls without a priori assumptions about object properties, such as friction or geometry.**

*Index Terms*—**Tactile Sensing, In-Hand Manipulation.**

## I. INTRODUCTION

Robotic within-hand manipulation [1] affords robot systems to manipulate objects in tight spaces and avoid gross arm movements, a particularly useful ability in cluttered or constrained environments. However, due to uncertainties in object properties, like friction, successful reorientations can prove to be a challenging task. Some approaches have used inverse kinematics with a highly constrained rigid hand and taking advantage of overcoming friction during sliding to reorient an object [2], while others have taken advantage of compliant or under-actuated systems [3]. However, controlling for object slip directly, without such models, can enable a much faster reorientations with unknown objects, an important feature in situations that necessitate faster response time such as in assembly lines or active disaster zones.

Thus far, aggressive dynamic manipulation has been accomplished using learned control policies, whether exploring real-world object contacts [4] or in simulation [5]. However, using a nibbed capacitive tactile sensor developed by Huh et al. [6] (Fig. 1) we hope to demonstrate that dynamic manipulations can be performed using simple control policies by only training for object motion recognition, thus making the sensor more generalizable to different scenarios while reducing the need for complex computing.

In this letter we explore the sensor's ability to detect speed

Y. Gloumakov, T. Huh, and H. Stuart are with the Mechanical Engineering Department, University of California, Berkeley, CA 06511 USA, (email: {yurigloum, thuh, hstuart} @berkeley.edu).

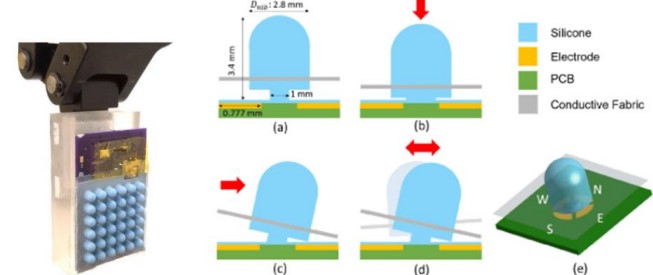

Figure 1. On the left, the sensor can be seen mounted on the tip of a robotic finger. The tactile sensor is made up of a grid of nibs according to dimensions in (a), where the deflection of each nib is tracked in 4 directions. These deflections are used to track pressure (b), sheer (c), and vibrations (d) that can be used detect slipping. The conductive fabric that is embedded in the nibs and deflected changes the capacitive signal between itself and the electrodes. Figure images were borrowed from [6].

of a slipping object as it slides across the sensor. While incipient slip has been demonstrated in various systems [7], [8], slip detection and regrasping can be leveraged to quickly reposition an object within the hand with minimal arm or finger movement [9], [10]. Meanwhile, steady-state slipping speed has only been demonstrated when objects are either much smaller than the sensor or not making contact with its entire surface [11], [12], so that the geometry or forces of an edge contact can be tracked over time. However, objects in a factory setting or during sorting are often fully flush and flat with the sensor and controlling the slip is necessary for dynamic manipulation. We hypothesize that the deflection of the sensor's nib interface would undergo a stick-slip interaction yielding characteristic frequencies and deflection amplitudes unique to each combination of material and slip speed.

## II. METHODS

To discover how the sensor detects slipping speed, we created a testbed that allowed us to test different slipping speeds and materials. The testbed was designed to maintain a constant distance between the sensor and a sliding object (Fig. 2); keeping the pressure constant was another consideration. Three rectangular objects made of different materials were tested: cherry, basswood, and acrylic with dimensions of 200x40x3 mm. The objects were pulled 134 mm by a string attached to a UR-10 robotic arm. The objects were then pushed back to the starting point and pulled again while sensor data was recorded

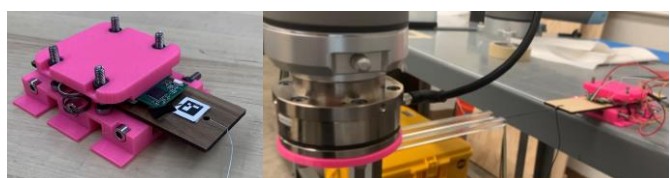

Figure 2. The left figure depicts the testbed that hosts the sensor and allows the object to slide through, rolling over a set of smooth bearings. On the right the robot arm can be seen to pull on the object by a string. The acrylic piece is placed on the end effect to push the object back into place.

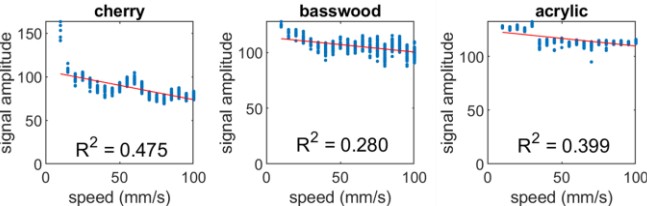

Figure 4. The raw signal amplitude is plotted against the pull speed for each of the three materials. The average amplitude during each pull cycle is plotted as a single point. A linear regression fit is overlaid.

at 600 Hz. This push-pull cycle lasted for 2 minutes for each speed setting, and speeds were varied from 10-100 mm/s in 5 mm/s increments. Since only the steady-state speed regime was of interest, the data from the acceleration and deceleration were spliced out. The termination of acceleration and initiation of deceleration were estimated to occur within the first 1/8th and the last 1/6th of the slipping period, respectively, with a conservative margin.

A feature of the nibbed sensor is its Programmable System on Chip (PSoC) infrastructure that enables us to couple any desired set of electrodes that result in a faster signal at the cost of resolution. Because we constrained the slip to a single linear direction, the nib deflection only needed to be tracked along a single axis (Fig. 3). Using a fast Fourier transform (FFT) the signal was converted into the frequency spectrum. Linear regressions are used to create a model using both the amplitude signal and the frequency spectrum separately to discover a fit that could identify the speed and material properties from a new signal. To obtain the frequency spectrum, a 300-frame sliding window was used, with an overlap of 1 frame to maximize the amount of extracted data.

Due to steady state slipping, the frequency responses were regarded as independent samples. Here, a frequency sample is a vector of length *n*, which corresponds to 300 (half the sampling rate) divided by the bin size, varied from 1 to 300, and where vector values correspond to their respective frequency amplitudes. Both the frequency response, as well as the raw signal amplitude, were averaged during each pull cycle; this meant that during the 2-minute data collection, the slower speed trials yielded fewer cycles and therefore less data. The data was used in building a regression and exploring classification and clustering methods.

### III. RESULTS

An example of amplitude data during one of the trials is shown in figure 3. At the lowest speed, over the course of 2

minutes, only 7 pull cycles were collected, while at the fastest speed, up to 44 cycles were collected over the course of the same period. The mean amplitude of each cycle is plotted in figure 4. The linear fit $R^2$ values were 0.475, 0.280, and 0.399 cherry, basswood, and acrylic objects, respectively. Although this corresponds to weak correlations, at speeds below 50 mm/s the correlation appears stronger.

In the frequency domain, linear fits have weaker correlations still when looking at individual frequency bins. In figure 5, we explore the correlation between speed and frequency bands, which consisted of the signal across any number of frequency bins simultaneously; in the figure only the highest and lowest correlations are displayed. Only weak correlation persisted.

### IV. DISCUSSION

In this work we observed that neither signal amplitude nor frequency responses yielded a strong correlation. Nevertheless, a negative correlation persisted, suggesting that there is an exploitable relationship which can be used to identify the speed at which an object is slipping. However, at speeds below 50 mm/s, a stronger relationship can be seen, and therefore, this would likely be the region that should be explored further in future data collections. This was not an unexpected result, as the difference between speeds was likely to plateau above a critical speed; nibs experience shorter stick times with increasing substrate speed, likely leading to a saturation in the amplitude signal [13]. Additionally, it appears there are differences in the amplitude response between materials that we believe can be used to train a classifier.

The results related to raw amplitude signal can be seen to have a sinusoidal feature over speeds. We suspect that this corresponds to a resonant frequency related to the testbed. Alternatively, this could be due to the nonlinearity of the robotic arm as it moves in a straight line.

Although the raw amplitude signals display a correlation with speed, it is highly susceptible to changes in grasp force, a

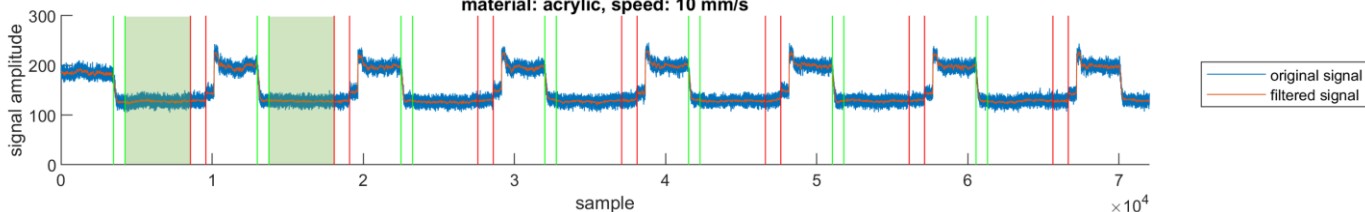

Figure 3. Example of a 2-minute push-pull cycle is shown for a single trial. Initiation and termination of the of the pull corresponds to the first green and the second red vertical line pairs. Accelerations and decelerations are spliced out, therefore only the region between second green and the first red vertical line pairs is considered (highlighted region is shown for the first two pull cycles). The filtered signal is displayed for reference only. A brief pause in motion can be seen immediately after the second vertical red line, then a brief high amplitude signal generated by the object being pushed back to its starting point (the highest amplitude signal), and finally followed by a prolonged pause corresponding to a re-tensioning of the string.

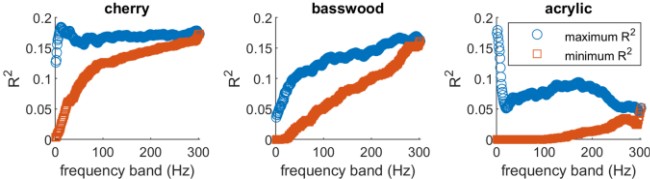

Figure 5. The maximum and minimum R2 value of the linear fit for each frequency band is displayed; these correspond to the highest and lowest correlations between specific frequency bands and slip speed. These values converge when the whole frequency spectrum is considered simultaneously, since there is only one frequency band.

factor that we deliberately accounted for by holding the distance constant. In an active controller, a sufficient grasp controller would need to be implement. However, frequency responses are less susceptible to grasp force, and the observation that certain frequency bands appear to find a correlation between the signal and sliding speed suggests that this would be a more reliable metric. Some short frequency bands appear to have very little correlations with speed, while others have a correlation. Out of all the tested frequency bins for the basswood and acrylic materials, 99.73% and 92.58%, respectively, exhibit a positive linear relationship between frequency bin amplitude and speed, while for the cherry material 100% of the tested frequency bins have a negative relationship. This suggests that material can likewise be determined by analyzing the frequency response.

Follow up work will include implementing classifiers that are capable of precisely distinguishing between materials and slipping speeds using, likely, the frequency signals. Ultimately, we hope to build a model capable of interpolating the data and identifying the speed with higher precision.

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
