# OpenReview forum: "Learning Slip with a Patterned Capacitive Tactile Sensor"
_ICRA.org/2022/Workshop/Contact-Rich — ICRA 2022 Workshop: RL for Manipulation Poster_

### Official Review · Reviewer_bwo1 · 2022-05-09
**Review of Paper "Learning Slip with a Patterned Capacitive Tactile Sensor"**

**Rating:** 6
**Confidence:** 3

**Review:**

Summary: This paper studies the problem of slip speed estimation of a grasped object from a tactile sensor. The work performs linear regression to determine the relationship between the amplitude (or frequency response) of the measured signal and slip speed.

Comments:

- The paper lacks information regarding what was the signal as depicted in Fig. 3. Assuming that the signal is proportional to the force along the sliding direction, this force would be the friction force, which has a non-linear relationship with velocity. This could explain the weak correlation between the signal amplitude and the slip speed.
- The results show weak correlations, so a natural extension would be to explore the use of multi-dimensional signals (e.g. obtained from different electrodes) instead of 1-dimensional.

---

### Official Review · Reviewer_4VBZ · 2022-05-09
**Good exploration for learning slip with high-frequency capacitive tactile sensors**

**Rating:** 5
**Confidence:** 3

**Review:**

Summary: This work explored predicting the slipping speed using a capacitive-based nib-structure tactile sensor. The authors designed a testbed for collecting controlled slips with different speeds and 3 different materials. The collected high-frequency signals are processed to fit a linear regression model to predict the slipping speed. The experiments show some weak correlation between the features and slipping speed.

Pros:
- Overall, it is clearly written and the experiment design is neat
- It is worth studying to use of high-frequency tactile signals for reactive/dynamic manipulation for future reference

Cons:
- The experiments did not show a strong correlation between the extracted features and the slipping speed.
- The extracted features were relatively simple. The authors could consider using more advanced machine learning models.